# Policy addressing suicidality in children and young people: a scoping review protocol

Lynne Gilmour, Edward Duncan, Margaret Maxwell

## ABSTRACT

**Introduction** Suicide is one of the leading causes of death among children and young people globally and a major public health issue. Government policies determine how much recognised health issues are prioritised and set the context for investment, development and delivery of services. A review of policies concerning children and young people who are suicidal could shed light on the extent that this public health issue is prioritised and highlight examples of best practice in this area. There has never been a review to explore how policy worldwide addresses the specific needs of this vulnerable population. This review will map the key policy documents and identify their relevance to the review question: how does policy address the treatment and care of children and young people who experience suicidality? (international, national (UK) and local (Scotland)).

**Methodology** Employing scoping review methodological guidance a systematic and transparent approach will be taken. Preliminary searches will facilitate the identification of MeSh terms, subject headings, individual database and platform nuances. A full search strategy will be created to search five databases: CINAHL, PsychInfo, Medline, Web of Science and Cochrane. Government and other key agency websites (eg, WHO, Unicef) will be searched to identify policy documents. The reference lists of identified documents will be checked. A second reviewer will independently screen and cross validate eligible studies for final inclusion. A data extraction template will then be used to extract key information. We will report our findings using narrative synthesis and tabulate findings, by agreed key components.

**Ethics and dissemination** Ethical approval is not required to conduct a scoping review. We will disseminate the findings through a peer-reviewed publication and conference presentation.

NMAPH Research Unit, University of Stirling, Stirling, UK

**Correspondence to**
Lynne Gilmour; Lyg1@stir.ac.uk

## Strengths and limitations of this study

► This will be the first scoping review of policy to consider how policy addresses suicidality in CYP (children and young people), mapping the relevant documents, as well as identifying any gaps.
► Using a systematic approach and methodological guidance, this review will be rigorous and credible.
► The review will be limited in that it will only include documents written in English.
► Given the iterative nature of searching for policy documents, this review will not capture every available worldwide policy pertaining to suicidal children and young people but will situate Scottish and UK policy within an international context.

## INTRODUCTION

Reducing suicide rates is a global public health priority and a specific target of the first WHO Mental Health Action Plan[1] which called for a lifespan approach to mental health. Member nations of the WHO were then called to implement national suicide prevention strategies.[2] To date, this has been achieved in 28 countries with most suicide prevention strategies covering the entire lifespan.[3]

Suicide is the second leading cause of death among children and young people worldwide,[4] and in the UK, is the main cause of death.[5] A recent UK report notes that 14% of deaths among 10–19 years old in England and Wales were caused by suicide and that having an undetermined cause of death recorded was much higher among under 16 years old meaning that potentially this number could be much higher.[5]

As suicide prevention strategies have taken a lifespan approach, there is potential for the specific needs of children and young people to become lost. A recent review highlighted that there were even differences between the presenting issues for children (aged <15 years) compared with adolescents (not defined) who later died by suicide.[6] It is widely acknowledged that children and young people have different needs to adults, and different health policy and service provision is common place in many countries. Yet, little is known about how policy specifically addresses suicidality in children and young people and which policies can serve as examples of best practice or innovation in tackling suicide in this population.

To date, most systematic reviews in this area have focused on the effectiveness of interventions, and not the content of policy documents themselves.[7–9] Government policies determine how much recognised health issues are prioritised and set the context for investment, development and delivery of services. However, the gulf that often exists between policies and practice is highlighted by Fortune and Clarkson[10] who state that child and adolescent mental health services in New Zealand are overwhelmed with referrals and lack the tools and resources to address suicide in young people beyond assessing risk.[10]

To develop or evaluate services and interventions for children and young people who are suicidal, the policy context within which they are delivered must be taken into account. Despite the development of strategy documents that aim to address suicide and children and young people being identified as a priority population, it is unclear how policy per se addresses the needs of suicidal children and young people specifically. How the vision expressed in suicide prevention policies translates into the separate government policies that are concerned with addressing the needs of children and young people who are suicidal within countries is also unclear.

This review will use scoping review methodology[11–13] to map the relevant available policies worldwide and establish how they address the treatment and care of children and young people who are suicidal. The review will enable gaps in policy provision for children and young people who are suicidal to be identified and highlight the strategic direction for treating and responding to individual children and young people who have either attempted or considered suicide. This review will directly contribute to a larger researcher project that will be conducted in Scotland, concerned with child and adolescent mental health services treatment of children and young people who are suicidal.

## METHODOLOGY

The scoping review question—how does policy address the treatment and care of children and young people who experience suicidality? (international, national (UK) and local (Scotland)) was developed using the Joanna Briggs Institute guidance for umbrella reviews.[13] The search strategy will be systematic in its approach, employing established guidelines[11–13] to inform the methodological process.

### Review aims

► Identify and map policy documents that relate to children and young people who are suicidal.
► Determine how the policies relate to and address children and young people who are suicidal.
► Establish any gaps within the policies in terms of addressing children who are suicidal.
► Explore the potential need for any future thematic or discursive analysis of how policy deals with children and young people who are suicidal.

**Table 1** Keyword search terms

| Concept | Keywords |
|---|---|
| Children and young people (5–18 years) | Child*; 'young people'; youth; adolesc*; teen*; paediatric |
| Suicide | Suicide; suicidal; |
| Policy | Policy; Procedure; Guidance; Strategy |
| Limits | English Language; Published after 2000 |

### Search strategy

The review question was used to generate keywords that will be used as search terms (table 1).

These search terms will be amended for each of the different databases and platforms used to include MeSH terms and subject headings. Preliminary searches were conducted to refine the search terms and identify the most appropriate databases and platforms. Four databases will be searched: CINAHL; Medline; PsychInfo and the Cochrane Database of Systematic Reviews. The websites of the following key government, statutory and non-statutory agencies will be searched, focusing on postindustrial nations with developed economies in order to identify those with most applicability to the UK: WHO; Unicef, UK government; Scottish government; ScotPHO; UK National Institute for Health and Care Excellence (NICE); National Office of Suicide Prevention (Ireland); Ministry of Health NZ; Australian Government Website and the Mental Health Commission Canada. Google and Google Scholar will also be used to identify other policy documents and grey literature. References of identified documents will be checked. Leading experts in the field will be emailed and requested to list key policies as a method of triangulating data collection and ensuring rigour.

### Population

Key characteristics of the study population are age and suicidality. Defining this population is complicated by the variation in what age constitutes being a child or a young person and by definitions of self-harm to include suicidal intent. The United Nations Convention on the Rights of the Child states anyone under 18 years is a child, the WHO defines adolescence as age 12–18 years and young people as 12–24 years and the United Nations use the term 'youth' to refer to people from aged 12 into their 30s.[14] Given the context of the review is Scotland, and the Children's Scotland Act (1995)[15] defines a child as anyone under 18 years and the NICE guidelines[16] for self-harm relate to children over 8 years, it was agreed that these parameters would be used. Similarly, the use of definitions of self-harm to include suicide attempts, mean that some papers concerned with self-harm may be relevant to a review of policies relating to suicide. As the phenomenon of interest is suicide, documents that refer solely to non-suicidal self-injury will be excluded, and self-harm

will not be used as a search term. The target population of children and young people will include all suicidal children and young people who are aged 8–18 years, regardless of gender and ethnicity. Documents that relate solely to adult (>18 years) populations, or solely infant populations (<5 years), will be excluded. However, policies that have a generic title and age span will be included.

## Concept

This review is primarily concerned with establishing what policy documents say in relation to the treatment and care of this population, as well as highlighting any variations or gaps in policy. Prevention activity can be universal (eg, public health approach aimed at everyone), selected (targeted at high-risk groups) and indicated (at the individual treatment or intervention level). This review is concerned with identifying policy that includes indicated prevention activity: specifically, about the treatment and care for children and young people who are suicidal.

## Context

The reviewing authors are based in Scotland (UK), and this review will contribute to a larger study being conducted there. Although Scotland remains part of the UK, health has always been a devolved issue, and the National Health Service policy in Scotland and other regions of the United Kingdom is markedly different. The relevance and applicability of each policy document to Scotland and the UK, will be assessed and coded using the following criteria:
A. Directly relevant to Scotland
B. Directly relevant to the UK.
C. Includes non-UK studies, but the context/population group would apply equally to UK settings.
D. Includes non-UK studies that are clearly not relevant to UK settings.

## Types of sources

All relevant national and international policy documents will be included. Local policy documents will refer to Scotland. Individual regions and states may have they their own guidelines, as will individual organisations as part of their workforce policies and procedures, however, these should reflect the national priorities and guidance. Although there are subtle differences in defining the terms 'strategy' and 'policy', they are often used interchangeably, and some countries now also use the term 'framework' to outline a national approach. For the purposes of this review, policy documents can include policies, policy guidance, strategies, codes of conduct, national service frameworks, practice guidance, white and green papers.[17] Reviews of policy documents centred on CYP who are suicidal will also be included to support what is known in this area.

## Screening

Identified policies and documents will be listed in an excel workbook and screened against agreed inclusion and exclusion criteria (table 2), first by title and executive summary or abstract (by two authors independently) and then in full text (by author one, with a random sample of 50% cross validated by a second author). Any disagreements will be discussed with a third author who will act as mediator. All discussion and agreements will be recorded. A Preferred Reporting Items for Systematic Reviews and Meta-Analyses diagram[18] will be used to record and illustrate the search process.

## Data extraction

A data extraction template will be developed that reflects the research question and the aims of the review. It will include gathering information regarding the key characteristics of the policy document, such as year, country, type

| Table 2 Search criteria | |
| --- | --- |
| **Inclusion** | **Exclusion** |
| About children <18 years and >8 years. | Not in English |
| About suicide OR uses a definition of self-harm to include suicidal behaviour. | Published before 2000 |
| National policy documents/strategies | Solely about non-suicidal self-injury |
| About national policy/strategy documents | Solely about universal and selective prevention |
| Reviews of policy | Solely about a population of adults >18 |
| Most recent version of policy document or strategy | Solely about a population of children <8 years |
| Directly relevant to Scotland only OR Relevant to the UK OR includes non-UK studies, but the context/population group would apply equally to UK settings | Primary studies |
| | Includes non-UK studies that are clearly not relevant to UK settings. |
| | Organisational policies for example, at service level. |
| | Previous version of policy where newer version is available for inclusion. |

of policy etc as well as specifics about the policy approach and detail of how it pertains to suicidal children and young people. One reviewer (LG) will complete data extraction, with a second reviewer cross validating through a process of independently extracting data (25%) from a random sample of included policies. Any disagreements or inconsistencies in the extracted data will be resolved via discussion and will involve the third reviewer.

## PATIENT AND PUBLIC INVOLVEMENT

The public and patients were not consulted during the development of this scoping review protocol.

## ETHICS AND DISSEMINATION
### Charting the data and presenting the findings

Review findings will be presented in a summary table with headings that will reflect the research question and key concepts that emerge, organised by categories such as country and type of document. Scoping reviews are used to provide a broad overview rather than in-depth analysis of a topic area and can often help to establish the feasibility of a future systematic review or qualitative evidence synthesis. Narrative will be used to provide a descriptive overview of the included policy and research and indicate any identified gaps. Lay and executive summaries will be produced to make the findings relevant and accessible to the public, practitioners and policy makers. The findings will be widely disseminated through a peer-reviewed publication and conference presentations.

Suicide among children and young people is a global public health concern. This review will be the first to identify and map international policy pertaining to the treatment and care of CYP who are suicidal.

Acknowledging the iterative process involved in identifying policy documents and potential language barriers, this review will not capture every policy pertaining to suicidal children and young people across the globe. However, by applying a systematic and well-defined approach, this review will robustly and reliably summarise and map the key policy documents. It will then describe how these policies address or perhaps do not address the treatment and care needs of children and young people who are suicidal. It will provide invaluable knowledge for future policy makers, researchers and practitioners.

**Acknowledgements** Contribution was made by Maya Jeffery, Faculty of Health Sciences Librarian at the University of Stirling, who assisted in the identification of key search terms and appropriate databases. LG led on development of the protocol and all authors contributed to writing of the paper.

**Contributors** LG: the idea for the protocol was conceived. LG, ED, MM: the search strategy and parameters of the review were established and agreed equally. LG: the protocol was first drafted. ED, MM: it was then edited and redrafted. LG, ED, MM:

the manuscript was then edited and redrafted following reviewers' comments. All authors have contributed to and approved the final version of the manuscript.

**Funding** This research is funded by Economic and Social Research Council UK (ESRC)

**Competing interests** None declared.

**Patient consent** Not requried.

**Ethics approval** No ethical approval is required to conduct a review.

**Provenance and peer review** Not commissioned; externally peer reviewed.

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
