## [Reviewer comments · BMJ Open]

ARTICLE DETAILS

TITLE (PROVISIONAL)	Policy addressing suicidality in children and young people: a scoping review protocol.
AUTHORS	Gilmour, Lynne; Duncan, Edward; Maxwell, Margaret

VERSION 1 – REVIEW

REVIEWER	Charlotte Connor University of Warwick
REVIEW RETURNED	09-Apr-2018

GENERAL COMMENTS	A timely and important paper. I look forward to reading results
---

REVIEWER	Paul Tiffin University of York, UK
REVIEW RETURNED	06-Jun-2018

GENERAL COMMENTS	It is a timely and useful review. The definition of a young person is sensible for UK purposes. The background may benefit from some expansion, for example citing some prevalence rates for suicide in under 18s. The search strategy seems appropriate. However, in practice it may be difficult to separate out policy concerned with over and under 18s. There is a risk that general policies, which make some mention of young people, may be missed and it would be important to know how the authors may address this issue. The authors should also state why they will use a narrative synthesis rather than apply, say, a thematic analysis, though the former does seem most appropriate for a scoping review. Minor points: 1. The phrase 'Scottish and UK policy' reads slightly awkwardly, as, so far, Scotland is still part of the UK.2. The acronym 'CYP' is used before it is spelled out in full. Also, it is probably unnecessary to use.3. I presume the 50% cross validation sample will be randomly sampled? If so this should be stated.4. "ACKNOWLEDGEMENTS" is misspelled.
--

VERSION 1 – AUTHOR RESPONSE

Response to Reviewers Comments

Comment	Response
---------	----------

“The background may benefit from some expansion, for example citing some prevalence rates for suicide in under 18s.”	 • The Introduction on p3 now reads “Suicide is the second leading cause of death amongst children and young people worldwide (7), and in the UK is the main cause (8) A recent UK report notes that 14% of deaths amongst 10-19yr old’s in England and Wales were caused by suicide, and that having an undetermined cause of death recorded was much higher amongst under 16yr old’s meaning that this statistic could be potentially higher (8). As suicide prevention strategies have taken a lifespan approach, there is potential for the specific needs of children and young people to become lost.”
However, in practice it may be difficult to separate out policy concerned with over and under 18s. There is a risk that general policies, which make some mention of young people, may be missed and it would be important to know how the authors may address this issue.	 • The authors acknowledge that many policies may be generic and under the sub-heading population on p6. have noted “The target population of children and young people will include all suicidal children and young people who are aged 8-18 years, regardless of gender and ethnicity. Documents that relate solely to adult (>18 years) populations, or solely infant populations (< 5 years), will be excluded. However, policies that have a generic title and age span will be included. • Also, under Data Extraction on p8 & 9 the authors explain that they will extract from the policy documents content that relates to children and young people who are suicidal. “It will include gathering information regarding the key-characteristics of the policy document, such as year, country, type of policy etc. as well as specifics about the policy approach and detail of how it pertains to suicidal children and young people.” • The authors acknowledge that the review may miss some relevant policies and under Limitations on p9 write “Acknowledging the iterative process involved in identifying policy documents, and potential language barriers, this review will not capture every policy pertaining to suicidal children and young people across the globe. However, by applying a systematic and well-defined approach this review will robustly and reliably summarise and map the key policy documents.”
The authors should also state why they will use a narrative synthesis rather than apply, say, a thematic analysis, though the former does seem	 • P5. now includes as an aim of the research “Explore the potential need for any future thematic or discursive analysis of how policy deals with children and young people who are suicidal.”

most appropriate for a scoping review.	 • P9 now includes a justification for narrative synthesis. “Scoping reviews are used to provide a broad overview rather than in-depth analysis of a topic area and can often help to establish the feasibility of a future systematic review or qualitative evidence synthesis. Narrative will be used to provide a descriptive overview of the included policy and research and indicate any identified gaps.”
The phrase ‘Scottish and UK policy’ reads slightly awkwardly, as, so far, Scotland is still part of the UK.	 • P7 under the sub-heading Context now includes “Although Scotland remains part of the UK, health has always been a devolved issue and the NHS policy across the two nations is markedly different”
The acronym ‘CYP’ is used before it is spelled out in full. Also, it is probably unnecessary to use.	 • The acronym CYP has now been replaced throughout the manuscript with the words children and young people in full.
I presume the 50% cross validation sample will be randomly sampled? If so this should be stated.	 • In relation to Screening p8 now reads “...with a random sample of 50% cross validated by a second author)”
“ACKNOWELDGEEMENTS” is misspelled.	 • P10 now reads “ACKNOWLEDGEMENTS”